# A Reconstructed Human Melanoma-in-Skin Model to Study Immune Modulatory and Angiogenic Mechanisms Facilitating Initial Melanoma Growth and Invasion

**DOI:** 10.3390/cancers15102849

**Published:** 2023-05-20

**Authors:** Elisabetta Michielon, Marta López González, Dorian A. Stolk, Joeke G. C. Stolwijk, Sanne Roffel, Taco Waaijman, Sinéad M. Lougheed, Tanja D. de Gruijl, Susan Gibbs

**Affiliations:** 1Department of Molecular Cell Biology and Immunology, Amsterdam UMC, Location Vrije Universiteit, De Boelelaan 1117, 1081 HV Amsterdam, The Netherlands; 2Amsterdam Institute for Infection and Immunity, 1105 AZ Amsterdam, The Netherlandstd.degruijl@amsterdamumc.nl (T.D.d.G.); 3Cancer Center Amsterdam, Cancer Biology and Immunology, 1081 HV Amsterdam, The Netherlands; 4Department of Medical Oncology, Amsterdam UMC, Location Vrije Universiteit, 1105 AZ Amsterdam, The Netherlands; 5Department of Oral Cell Biology, Academic Centre for Dentistry Amsterdam (ACTA), University of Amsterdam and Vrije Universiteit, 1105 AZ Amsterdam, The Netherlands

**Keywords:** tumor progression, reconstructed human skin, melanoma, immune modulation, endothelial cell sprouting

## Abstract

**Simple Summary:**

The melanoma-conditioned microenvironment promotes immune escape and tumor progression, contributing to resistance to anti-melanoma immunotherapy in a large group of treated patients. While two-dimensional cultures lack tissue-context, animals poorly predict human immune responses. Thus, a clear need exists for more physiological human models. We previously described an in vitro human melanoma-in-skin (Mel-RhS) model with the SK-MEL-28 cell line mirroring features of invasive melanoma. Here, we further investigated the tumor-induced microenvironment of this model, generated five additional Mel-RhS models (A375, COLO829, G361, MeWo, and RPMI-7951), and performed in-depth analysis of invasion, immune modulation, angiogenesis, and their respective mediators. We present three-dimensional models allowing for the study of diverse tumor-intrinsic properties that could be used to investigate efficacy of therapeutic interventions. By mimicking different stages of melanoma and related features, each Mel-RhS can be further individually tailored to obtain a physiologically evermore relevant model to study specific mechanisms underlying melanomagenesis and progression.

**Abstract:**

Invasion, immune modulation, and angiogenesis are crucial in melanoma progression. Studies based on animals or two-dimensional cultures poorly recapitulate the tumor-microenvironmental cross-talk found in humans. This highlights a need for more physiological human models to better study melanoma features. Here, six melanoma cell lines (A375, COLO829, G361, MeWo, RPMI-7951, and SK-MEL-28) were used to generate an in vitro three-dimensional human melanoma-in-skin (Mel-RhS) model and were compared in terms of dermal invasion and immune modulatory and pro-angiogenic capabilities. A375 displayed the most invasive phenotype by clearly expanding into the dermal compartment, whereas COLO829, G361, MeWo, and SK-MEL-28 recapitulated to different extent the initial stages of melanoma invasion. No nest formation was observed for RPMI-7951. Notably, the integration of A375 and SK-MEL-28 cells into the model resulted in an increased secretion of immune modulatory factors (e.g., M-CSF, IL-10, and TGFβ) and pro-angiogenic factors (e.g., Flt-1 and VEGF). Mel-RhS-derived supernatants induced endothelial cell sprouting in vitro. In addition, observed A375-RhS tissue contraction was correlated to increased TGFβ release and α-SMA expression, all indicative of differentiation of fibroblasts into cancer-associated fibroblast-like cells and reminiscent of epithelial-to-mesenchymal transition, consistent with A375′s most prominent invasive behavior. In conclusion, we successfully generated several Mel-RhS models mimicking different stages of melanoma progression, which can be further tailored for future studies to investigate individual aspects of the disease and serve as three-dimensional models to assess efficacy of therapeutic strategies.

## 1. Introduction

Melanoma is the most aggressive form of skin cancer that arises from melanocytes, specialized pigmented cells protecting the skin against UV-radiation. Mutations in melanocyte DNA can lead to loss of control of key growth regulatory genes, secretion of autocrine growth factors, and downregulation of adhesion molecules, which in turn disrupt cell homeostasis [1]. This results in continuous melanocyte proliferation and can eventually lead to the formation of a naevus [2]. Although naevi are generally benign, they constitute risk factors for subsequent melanoma formation upon additional mutagenic hits. In its initial stages, also known as the radial growth phase (RGP), where cells are confined to the epidermal layer, melanomas can be removed via surgical excision [3,4]. With the progression of the disease, melanoma cells can enter the vertical growth phase (VGP), which is characterized by cell nests invading the dermis, that can advance to metastatic malignant melanoma with cancer cells migrating into the blood or lymphatic circulation through which they can reach distant organs [4].

Immune modulation plays a key role in melanoma progression and dissemination. Secretion of cytokines, such as anti-inflammatory IL-10 and transforming growth factor β (TGFβ), is involved in the establishment of a tumor-favorable immune suppressive microenvironment [5]. This is essential for enabling melanoma cell survival, immune escape, and thus cancer progression, and is achieved in large part by inhibiting the differentiation of myeloid cell precursors into immune stimulatory subsets, e.g., dendritic cells (DCs) and M1 macrophages, while promoting their differentiation into regulatory subsets, such as myeloid-derived suppressor cells (MDSCs) and M2 macrophages [6,7]. Melanoma-secreted TGFβ also leads to the transformation of normal fibroblasts into pro-tumorigenic cancer-associated fibroblasts (CAFs), which share features with myofibroblasts that are active during wound healing processes, expressing alpha-smooth muscle actin (α-SMA) and showing increased contractility [8]. CAFs are able to further shape the tumor-favoring milieu by affecting the recruitment and function of various innate and adaptive immune cells. This occurs by supporting tumor angiogenesis via the secretion of different vessel formation promoting factors (e.g., vascular endothelial growth factor, VEGF) and impairs melanoma susceptibility to immunotherapy. Altogether, these processes support all phases of melanomagenesis, from tumor development to the metastatic cascade [9].

The formation and growth of new vessels are also essential features in the metastatic process by providing nutrients as well as providing the principal route for the entering of tumor cells into the blood circulation and ultimately resulting in tumor cell extravasation and metastasis [10]. In vivo, melanoma cells secrete a plethora of pro-angiogenic factors, including VEGF, basic fibroblast growth factor (bFGF or FGF-2), interleukin-8 (IL-8), placental growth factor (PlGF), and platelet-derived growth factor (PDGF) [11]. The biological functions of these soluble factors have been evaluated in both in vitro tumor angiogenesis models as well as in xenograft mouse models [12,13]. The metastatic process is characterized by the dynamic interplay between immune suppression, transformation of fibroblasts into CAFs, and angiogenesis, all of which are promoted by diverse soluble factors.

Most studies reporting to date on the relationship between the tumor and its microenvironment have relied on animal models or on classical two-dimensional (2D) cultures, which poorly mimic the environmental cross-talk in humans. To better study melanoma features in a human setting and to overcome inherent limitations of both adherent 2D cell cultures and in vivo animal studies in reflecting the human pathophysiology, three-dimensional (3D) in vitro organotypic human melanoma models have been developed [14,15,16,17,18,19,20,21,22,23,24]. We have previously described a full-thickness melanoma reconstructed human skin (Mel-RhS) model, based on the SK-MEL-28 melanoma cell line, recapitulating early melanoma invasive features and showcasing immune suppressive properties via the secretion of IL-10 in culture supernatants [15]. Here, the aim was to investigate melanoma cell growth, pro-angiogenic features, mesenchymal transformation, and immune modulation potential of five additional melanoma cell lines (A375, COLO829, G361, MeWo, and RPMI-7951) in the 3D tumor microenvironment (TME) and establish relevant models for more in-depth study of these different cellular processes central to melanoma progression.

## 2. Materials and Methods

### 2.1. Blood and Tissue Collection

Peripheral blood was collected from healthy adult donors (Sanquin Blood Supply Services, Amsterdam, The Netherlands). Human foreskin was obtained from healthy donors (<6 years of age) undergoing circumcision, after obtaining informed consent from legal guardians, and used anonymously. Formalin-Fixed Paraffin-Embedded (FFPE) melanoma patient-derived tissue samples were acquired from the Pathology Biobank, part of the VU University Medical Center Biobank. Histopathology of in situ and invasive melanoma tissue samples was confirmed by an experienced pathologist.

### 2.2. Cell Isolation and Culture

#### 2.2.1. Primary Skin Cells

Epidermal cells (keratinocytes and melanocytes) and dermal fibroblasts were isolated from foreskins and cultured as previously described [25,26]. Epidermal cells were co-cultured in Dulbecco’s Modified Eagle Medium (DMEM; Lonza, Verviers, Belgium)/Ham’s F-12 (Gibco, Grand Island, NY, USA) in a 3:1 ratio containing 1% penicillin/streptomycin (P/S; Invitrogen, Paisley, UK), 1% UltroserG (UG; BioSepra S.A., Cergy, France), 0.1 μM insulin (Sigma-Aldrich, St. Louis, MO, USA), 1 μM hydrocortisone (Sigma-Aldrich), 1 μM isoproterenol (Sigma-Aldrich), and freshly supplemented 2 ng/mL keratinocyte growth factor (KGF; Sigma-Aldrich) at 37 °C and 7.5% CO_2_. Dermal fibroblasts were cultured in DMEM with 1% P/S and 1% UG at 37 °C and 5% CO_2_. Cells from up to passage 2 were used in the experiments.

#### 2.2.2. Melanoma Cell Lines

The melanoma cell lines A375, COLO829, G361, MeWo, RPMI-7951, and SK-MEL-28 were purchased from the suppliers displayed in Table 1. Table 1 also lists mutational status and anatomical origins of the used cell lines. The melanoma cell line SK-MEL-28 was cultured in DMEM supplemented with 1% P/S and 2% UG. A375 cells were cultured in DMEM supplemented with 10% fetal bovine serum (FBS; Corning, Corning, NY, USA) and 1% P/S. COLO829 cells were cultured in RPMI-1640 medium with 10% FBS and 1% P/S. MeWo and RPMI-7951 cells were cultured in ATCC-formulated Eagle’s Minimum Essential Medium (EMEM) supplemented with 10% FBS and 1% P/S. G361 cells were cultured in ATCC-formulated McCoy’s 5a medium with 10% FBS and 1% P/S. All cell lines were cultured at 37 °C and 5% CO_2_.

#### 2.2.3. Endothelial Cells

Dermal endothelial cells (ECs) from two different donors were isolated, purified, and cultured as previously described [26,32]. Briefly, ECs were cultured at 37 °C and 5% CO_2_ on gelatin-coated culture flasks in M199 medium (Lonza) containing 1% P/S, 2 mM L-glutamin, 10% heat-inactivated new born calf serum (h.i. NBCS; Invitrogen, Waltham, MA, USA), 10% human serum (HS; Sanquin, Amsterdam, The Netherlands), 5 U/mL heparin (Pharmacy VUmc, Amsterdam, The Netherlands), and 3.75 μg/mL endothelial cell growth factor (ECGF), isolated from crude extract from bovine brain (Department of Physiology, Amsterdam UMC, Amsterdam, The Netherlands). This combined medium is referred to as human microvascular endothelial cell (hMVEC) medium and was supplemented with 2.5 ng/mL VEGF and 2.5 ng/mL bFGF before use. Characterization of dermal ECs was performed by flow cytometry (Appendix A).

#### 2.2.4. Monocytes

CD14^+^ monocytes were isolated and selected via magnetic activated cell sorting (MACS) from peripheral blood mononuclear cells (PBMCs) as previously described [15]. Monocytes were cultured in RPMI-1640 medium (Lonza) with HEPES and l-glutamine (BioWhittaker, Lonza, Basel, Switzerland) supplemented with 10% h.i. fetal calf serum (FCS; HyClone, GE Healthcare, Chicago, IL, USA), 50 µM β-mercaptoethanol (2ME; Gibco), 100 IU/mL sodium-penicillin (Gibco), 100 µg/mL streptomycin (Gibco), and 2 mM l-glutamine (Gibco).

### 2.3. Construction of Reconstructed Human Skin with or without Melanoma Cells

RhS and Mel-RhS were constructed essentially as previously described [15,33] in 24 mm transwell plates (pore size of 8 μm; Corning). Briefly, Mel-RhS models were created by seeding 2.5 × 10^4^ cells of one of the melanoma cell lines onto the reconstructed dermal compartment two hours prior to epidermal cell seeding. After culturing for 4 days in submerged conditions, RhS and Mel-RhS were subsequently cultured at the air–liquid interface for 4 weeks. During air-exposure, culture medium consisted of DMEM/Ham’s F-12 (3:1) supplemented with 1% P/S, 0.2% UG, 0.5 μM hydrocortisone, 1 μM isoproterenol, 0.1 μM insulin, 2 ng/mL KGF, 1 ng/mL epidermal growth factor (EGF; Sigma-Aldrich), 10 mM l-serine (Sigma-Aldrich), 10 µM l-carnitine (Sigma-Aldrich), 25 μM palmitic acid (Sigma-Aldrich), 7 μM arachidonic acid (Sigma-Aldrich), 15 μM linoleic acid (Sigma-Aldrich), 0.4 mM ascorbic acid (Sigma-Aldrich), and 1 µM vitamin E (Sigma-Aldrich), and was refreshed twice a week. Before harvesting, cultures were incubated overnight in the above mentioned medium, but in the absence of hydrocortisone. These 24 h-conditioned culture supernatants were collected and stored at −20 °C. Tissue sections were fixed overnight in 4% paraformaldehyde and prepared for histological and immunohistochemical analysis.

### 2.4. (Immuno)histochemistry

FFPE 5-μm-thick tissue sections were used for morphological (hematoxylin and eosin staining, H&E) and immunohistochemical analysis of Melan-A (1:100, clone A103, M7196, Dako, Glostrup, Denmark) and α-SMA (1:200, clone 1a4, M0851, Dako), as previously described [15]. No antigen retrieval step was performed to stain for α-SMA. Stained tissue sections were photographed using the Vectra Polaris automatic imaging system (Akoya Biosciences, Marlborough, MA, USA).

### 2.5. Measurement of RhS and Mel-RhS Contraction

Surface area was compared between RhS, A375-RhS, and SK-MEL-28-RhS (N = 3 independent experiments, each performed with an intra-experiment duplicate). After 4 weeks in air-exposed conditions, before harvesting, RhS and Mel-RhS were photographed with a Powershot G12 camera (Canon, Tokyo, Japan) and surface area was measured using ImageJ software (version 1.46r).

### 2.6. Measurement of Cytokine Secretion in Culture Supernatant

Cytokine secretion from the RhS and Mel-RhS models was measured in 24 h-conditioned culture supernatants. Levels of TGFβ, granulocyte-macrophage colony-stimulating factor (GM-CSF), and macrophage colony-stimulating factor (M-CSF) were assessed by enzyme-linked immunosorbent assay (ELISA) by means of the respective DuoSet ELISA kit (R&D Systems, Minneapolis, MN, USA), according to the manufacturer’s instructions. Secretion of CCL2, CCL5, CXCL10, IL-6, IL-8, and IL-10 was determined by flex Cytometric Bead Array (CBA) analysis (BD Biosciences, San Diego, CA, USA), according to the manufacturer’s protocol. The V-Plex Angiogenesis Panel 1 (human) Kit (MSD, Rockville, MD, USA) was used to measure concentrations of bFGF, Flt-1 (or VEGFR-1), PlGF, Tie-2, VEGF-A, VEGF-C, and VEGF-D, according to the manufacturer’s manual. Supernatants from at least three independent experiments, each with an intra-experiment duplicate, were used for ELISA, CBA, and MSD.

### 2.7. Sprouting Assay

In vitro endothelial tube formation was studied as previously described [32]. Briefly, fibrin matrices were prepared by the addition of 0.5 U/mL thrombin (EMD Millipore, Burlington, MA, USA) to a 3 mg/mL fibrinogen (Enzyme Research Laboratories, Leiden, The Netherlands) solution in M199 medium (Lonza). A volume of 300 μL was added to the wells of a 48-well plate. After polymerization, thrombin was inactivated by incubation with hMEC medium, consisting of M199 medium, 10% HS, 10% h.i. NBCS, 1% P/S, and 2 mM L-glutamin. Dermal ECs were seeded overnight on top of the gels at a confluent density of 6 × 10^4^ cells/cm^2^. To assess the angiogenic potential of the RhS and Mel-RhS models, ECs were stimulated with hMEC medium containing 2 ng/mL TNF-α and 30% RhS-, A375-RhS-, or SK-MEL-28-RhS-derived culture supernatants. After 2 days, the sprouts formed by ECs into the fibrin matrices were photographed using a Nikon Eclipse 80i microscope (Nikon, Tokyo, Japan) and analyzed with NIS-elements AR software 3.2 (Nikon).

### 2.8. Monocyte Exposure to RhS- and Mel-RhS-Derived Culture Supernatants

Monocyte-derived dendritic cell (moDC) cultures were exposed to culture supernatants from either RhS or Mel-RhS as previously described [15]. Briefly, 2 × 10^4^ monocytes were cultured for 6 days either in the presence or absence of 30% RhS-, A375-RhS-, or SK-MEL-28-RhS-derived culture supernatants in a flat bottom 96-well plate in complete RPMI-1640 medium supplemented with 1000 IU/mL recombinant human GM-CSF (Immunotech, Prague, Czech Republic) and 20 ng/mL recombinant human IL-4 (Strathmann Biotec, Hamburg, Germany). The same procedure was followed in the M-CSF, IL-10, and TGFβ blocking experiments, but supernatants from RhS and SK-MEL-28-RhS were pre-treated for 30 min with either anti-M-CSF (Tebu Bio, Le Perray, France), anti-IL-10 (clone 23738.11; Abcam, Cambridge, UK), anti-TGFβ (clone 1D11; R&D Systems), or IgG1 isotype (ICN Biomedicals, Irvine, CA, USA) as a control, all at 10 µg/mL.

### 2.9. Flow Cytometry

Culture supernatant-exposed monocyte-derived cells were harvested for fluorescence-activated cell sorting (FACS) analysis as previously described [15]. The following antibodies were used to assess surface marker expression: BDCA3-FITC (Miltenyi Biotec), CD1a-PE (BD Pharmingen, Franklin Lakes, NJ, USA), CD14-PerCPCy5.5 (BD Pharmingen), CD16-BV650 (BD Biosciences), CD163-BV421 (BD Horizon, Franklin Lakes, NJ, USA), PD-L1-APC (eBioscience, San Diego, CA, USA), PD-L2-BV711 (BD Horizon), and Fixable Viability Dye eFluor780 (eBioscience). Analyses were performed with Kaluza v.1.2.1 flow cytometry analysis software (Beckman Coulter, Brea, USA), FCS Express 6 (DeNovo Software, Glendale, CA, USA), or FlowJo v10 Software (BD Life Sciences, Franklin Lakes, NJ, USA).

### 2.10. Statistical Analysis

All data are presented as mean ± standard error of the mean (SEM). Statistical analysis was performed by means of either ordinary one-way ANOVA, Kruskal–Wallis test, paired *t*-test, or Wilcoxon test (Tukey post-hoc test) using GraphPad Prism 9 software (GraphPad Software Inc., La Jolla, San Diego, CA, USA). Differences were considered to be significant when *p* < 0.05.

## 3. Results

### 3.1. Melanoma Cell Lines Recapitulate Different Stages of the Disease in a 3D Human Melanoma-in-Skin Model

In the first stages of primary melanoma development, tumor cells are constrained to the epidermis (melanoma in situ), as visualized by H&E staining of a patient-derived melanoma (Figure 1a, left panel). During disease advancement, malignant cells are able to invade the basement membrane (BM) and spread vertically into the dermis (invasive melanoma), as indicated by the black arrow in Figure 1a.

The RhS and Mel-RhS models constructed with the different melanoma cell lines (Figure 1b,c) were compared with melanoma biopsies (Figure 1a). The control healthy RhS model consisted of a stratified epidermal layer on a fibroblast-populated fibrin-collagen hydrogel with Melan-A^+^ melanocytes distributed evenly throughout the basal cell layer (Figure 1b,c). The epidermis comprised a compact basal cell layer, stratum spinosum, stratum granulosum, and stratum corneum, in line with the epidermis of native healthy skin. Fibroblasts distributed throughout the hydrogel represented the dermis. The different melanoma cell lines grew and expanded differently when incorporated into the in vitro model, mimicking different stages of melanoma progression (Figure 1b,c). Introduction of RPMI-7951 cells into RhS did not lead to visible melanoma nest formation, whereas COLO829 resulted in sporadic small melanoma nests forming just below the epidermis, as indicated by the black arrows in Figure 1b,c. G361 cells formed many more of these nests in the upper part of the dermal layer compared to COLO829 (Figure 1b,c). The MeWo cell line expanded considerably at the dermal-epidermal interface, leading to a visibly thinner epidermis, as compared to the other models (Figure 1b,c). In line with our previous study [15], the SK-MEL-28-RhS physiologically resembled the initial stages of invasive melanoma, with melanoma aggregates observed growing into the dermis (Figure 1b,c). Amelanotic A375 melanoma cells showed the most extensive expansion and spreading, with melanoma nests present in the lower part of the reconstructed dermis (Figure 1b,c).

In conclusion, we generated Mel-RhS models reflecting different stages of melanoma growth, from the initial stage to the early metastatic/invasive stage.

### 3.2. Cytokine and Chemokine Release Profiles Differ between Mel-RhS Models

Tumors can hamper immune responses via multiple mechanisms including the secretion of immune modulatory and tissue remodeling cytokines and chemokines, and therefore supernatants of the different Mel-RhS models were analyzed for these factors. As shown in Figure 2, the A375-RhS, which displayed the most invasive tumor phenotype, exhibited an increased release of the pro-inflammatory chemokines and cytokines CCL2 (4-fold increase), GM-CSF (3-fold increase), IL-6 (7-fold increase), and IL-8 (9-fold increase) and of the anti-inflammatory cytokines IL-10 (229-fold increase), M-CSF (1.8-fold increase), and TGFβ (1.3-fold increase), when compared to the control RhS (Figure 2b). Despite their considerable melanoma cell expansion, MeWo-RhS only showed a trend towards an increase in M-CSF secretion (*p* = 0.0664), compared to the RhS control (Figure 2b).

In line with our previous publication [15], upregulated secretion of CXCL10 (5-fold increase), IL-10 (91-fold increase), and TGFβ (1.2-fold increase) in the SK-MEL-28-RhS-derived culture supernatants was detected compared to the RhS (Figure 2). Additionally, we identified higher secretion of CCL5 (6.7-fold increase) (Figure 2a) and GM-CSF (2.6-fold increase) (Figure 2b) in the SK-MEL-28-RhS model. No significant increase in cytokine secretion was found for the other Mel-RhS models (COLO829-RhS, G361-RhS, and RPMI-7951-RhS) compared to the RhS control (Appendix A and Appendix A), which is in line with their modest growth and spread in the in vitro model. Actual secreted cytokine concentrations for all Mel-RhS models are listed in Appendix A.

Overall, we can conclude that the different melanoma cell lines in the RhS model secreted unique profiles of immune modulatory cytokines and could therefore be expected to differentially modulate their tissue microenvironment.

### 3.3. SK-MEL-28-RhS and A375-RhS Suppress Monocyte-to-Dendritic Cell Differentiation through the Release of Soluble Factors: Relative Contributions of IL-10, M-CSF, and TGFβ

Due to the high release of immune modulatory cytokines from the A375-RhS and SK-MEL-28-RhS models (Figure 2), it was next determined whether culture supernatants from Mel-RhS constructed with these cell lines could interfere with the differentiation of monocytes into monocyte-derived dendritic cells (moDCs). When comparing the effects of the RhS supernatants versus the Mel-RhS supernatants, we found that monocytes cultured with the supernatants derived from either A375-RhS or SK-MEL-28-RhS adopted an M2-like phenotype (defined as CD14^+^CD163^+^CD16^+^) at a significantly higher frequency (Figure 3a). We previously reported that this immune suppressive effect of SK-MEL-28-RhS (i.e., skewing of T-cell stimulatory moDC differentiation to suppressive M2-like development) was in part mediated by IL-10 [15]. In the present study, we investigated which other factors might also be involved.

We constructed a correlation matrix heat map based on the levels of RhS- and Mel-RhS-secreted cytokines (in green in Figure 3b) and surface marker expression levels on monocytes (in black in Figure 3b). We observed co-clustering of M2-related markers (CD14, CD163, CD16), denoted by yellow boxes. In the case of SK-MEL-28-RhS, TGFβ, M-CSF, and IL-10 clustered most closely to this phenotype, while in A375-RhS, TGFβ co-clustered with the M2 markers, suggesting the possible involvement of these cytokines in the induction of the M2-like phenotype in the respective Mel-RhS models. Indeed, in the SK-MEL-28-RhS model, positive correlations were observed for IL-10, M-CSF, and TGFβ with the CD14^+^CD163^+^CD16^+^ M2-like phenotype and negative correlations for the same cytokines with the CD1a^+^ moDC phenotype (Figure 3b, black boxes in the left panel, all reaching significance). In the A375-RhS model, respective positive and negative correlations were found for TGFβ with the CD14^+^CD163^+^CD16^+^ M2-like and the CD1a^+^ moDC phenotype (Figure 3b, black boxes in the right panel), although these did not reach significance. Figure 3c shows these correlations for M-CSF and TGFβ in relation to CD1a^+^, CD14^+^, and M2-like populations for the SK-MEL-28-RhS model. These correlations were previously reported for IL-10 [15]. Interestingly, TGFβ showed the strongest correlation as compared to IL-10 and M-CSF (Figure 3b,c, Appendix A). Of note, secreted levels of TGFβ were also correlated with PD-L1 and PD-L2 expression levels (MFI) in the SK-MEL-28-RhS model (Appendix A).

In the A375-RhS model, significant correlations for M-CSF and TGFβ were mostly found in relation to BDCA3^+^ phenotypes (see Appendix A). BDCA3/CD141 was also previously related to a tumor-induced M2-like phenotype by us [15,34].

Because of their significant correlation with monocyte suppression induced by SK-MEL-28-RhS supernatants, we assessed the relative involvement of IL-10, M-CSF, and TGFβ in this process by adding neutralizing antibodies to the SK-MEL-28-RhS-conditioned monocyte differentiation cultures. A high-dimensional t-Distributed Stochastic Neighbor Embedding (t-SNE) analysis was performed based on the combined expression of BDCA3, CD14, CD80, CD1a, PD-L1, PD-L2, CD163, and CD16 on the monocytic population upon its exposure to supernatants from SK-MEL-28-RhS with either a control isotype (IgG1), or an M-CSF (Figure 4a) or a TGFβ (Figure 4c) blocking antibody. Similar results for IL-10 blockade have been previously reported [15]. Neutralization of both M-CSF (Figure 4a,b) and TGFβ (Figure 4c,d) resulted in a clear shift between different subpopulations within the conditioned monocyte population. For M-CSF blockade, this shift from one subpopulation to another (denoted by a magenta gate) was accompanied by a reduction in expression levels of BDCA3, PD-L1, and PD-L2 (Figure 4b). TGFβ blockade resulted in the partial loss of a subpopulation (denoted by the black gate) that was characterized by high expression levels of the typifying M2-like markers BDCA3, CD14, CD163, and CD16 as well as PD-L1 and PD-L2 (Figure 4d). In Figure 4e, the relative contributions of M-CSF, IL-10, and TGFβ to the SK-MEL-28-RhS-mediated monocyte suppression are shown by their individual or combined blockade. Significant and dominant suppression of BDCA3 expression by M-CSF as well as a dominant (though not significant) effect of TGFβ on the skewing from a CD1a^+^ to a CD14^+^ M2-like phenotype was apparent, while all three cytokines appeared to contribute to both PD-L1 and PD-L2 expression induced by SK-MEL-28-RhS.

### 3.4. A375-RhS and SK-MEL-28-RhS Induce Angiogenesis In Vitro

In general, the TME is characterized by angiogenesis, an essential feature to provide nutrients for tumor cell survival and growth [35]. Therefore, we studied whether the Mel-RhS models could also induce the formation of a pro-angiogenic microenvironment.

Secretion of pro-angiogenic factors from Mel-RhS was measured and compared to those from RhS (Figure 5). The presence of the A375 melanoma cells in the skin model led to an increased release of Flt-1 (2-fold increase), PlGF (2-fold increase), Tie-2 (1.8-fold increase), VEGF (1.8-fold increase), VEGF-C (1.7-fold increase), and VEGF-D (1.9-fold increase) (Figure 5a). Higher secretion levels of Flt-1 and VEGF were also found in the supernatants from MeWo-RhS (2-fold and 1.9-fold increase, respectively) and SK-MEL-28-RhS (1.7-fold and 1.5-fold increase, respectively) (Figure 5a). No increase in pro-angiogenic factors could be detected for COLO829-RhS, G361-RhS, and RPMI-7951-RhS) compared to the control RhS (Appendix A and Appendix A). Of note, secretion of pro-angiogenic factors from confluent monolayers of A375 and SK-MEL-28 melanoma cells was, for most of these factors, quite limited (Appendix A), in comparison to the release of the same factors in supernatants from the 3D melanoma models.

Given this higher secretion of pro-angiogenic factors by A375-RhS, MeWo-RhS, and SK-MEL-28-RhS, it was next investigated whether their secretome could induce angiogenesis in vitro over a 2-day period (Figure 5b,c). Sprouting of endothelial cells (ECs) was studied using 3D fibrin matrices cultured in the presence of 30% culture supernatants from RhS, A375-RhS, MeWo-RhS, or SK-MEL-28-RhS. Basal sprout formation could already be observed microscopically when ECs were exposed to RhS-derived supernatants, as indicated by the black arrows in Figure 5b. Surprisingly, no increased sprouting was found when ECs were exposed to MeWo-RhS-derived culture supernatants (the resulting mean sprouting area equaled 1, i.e., equaled the surface area observed with RhS SN), whereas a clear increase in sprout formation could be observed in EC cultures exposed to supernatants from both A375-RhS and SK-MEL-28-RhS (Figure 5b). Indeed, when the amount of sprouting was quantified as a percentage of the total surface area and normalized to the control RhS, sprouting surface area resulted in a 3.4-fold and 3.3-fold increase when ECs were exposed to supernatants from A375-RhS and SK-MEL-28-RhS, respectively (Figure 5c). Notably, no difference in sprouting was found between A375-RhS and SK-MEL-28-RhS indicating the equal potential of both melanoma cell lines to promote the formation of a pro-angiogenic microenvironment (Figure 5c).

To summarize, the presence of COLO829, G361, RPMI-7951, and MeWo cells in the skin model did not affect EC behavior, further strengthening the idea that these cell lines mimic very early stages of the disease. On the other hand, together with their immune modulatory capabilities, A375-RhS and SK-MEL-28 may reflect a more advanced stage of melanoma progression in terms of promoting the formation of a tumor-promoting and pro-angiogenic TME.

### 3.5. A375-RhS Represents a More Advanced Stage of Melanoma Progression and Induces Fibroblast Activation in a TGFβ-Dependent Fashion

CAF-induced mesenchymal transition is an important factor in promoting melanoma cell invasion into the underlying dermis and metastasis [36]. One of the most significant markers of fibroblast activation and CAF differentiation is the upregulation of α-SMA expression, which is a shared feature with myofibroblasts found in the wound healing and scarring processes [37,38,39]. We therefore investigated the expression of α-SMA in both A375-RhS and SK-MEL-28-RhS. Staining for this marker showed a clear population of α-SMA^+^ cells around the A375 and SK-MEL-28 nests (Figure 6a), suggesting a melanoma-induced transition of fibroblasts into myofibroblast-like cells in their proximity.

In addition, α-SMA expression was clearly higher in the A375-RhS and SK-MEL-28-RhS models, in comparison to that of RhS (Figure 6a), COLO829-RhS, G361-RhS, MeWo-RhS, or RPMI-7951-RhS (Appendix A). This result indicates a higher number of fibroblasts transitioning into CAF-like cells in the presence of the A375 and SK-MEL-28 melanoma cell lines and may thus reflect the more advanced invasive states of these cells within the in vitro models compared to the other studied melanoma cell lines.

CAFs also contribute to mechanical extracellular matrix remodeling through increased cell contractility. Interestingly, after 4 weeks of air-exposed culturing, a decreased surface area of the A375-RhS was observed, compared to both RhS and SK-MEL-28-RhS (Figure 6b,c), which might be related to the increased levels of myofibroblast-like cells surrounding the melanoma nests in the A375-RhS model. TGFβ has previously been shown to induce a contractile gene program that drives myofibroblast formation [40]. This was supported by a significant negative correlation between surface area and TGFβ secretion in culture supernatants from A375-RhS (Figure 3d). Although TGFβ levels were elevated in SK-MEL-28-RhS supernatants compared to RhS, they were lower than those observed for A375-RhS (see Figure 3d) and did not lead to enhanced contractility as evidenced by an unchanged surface area.

In general, CAFs aid melanoma progression and metastasis by secreting inflammatory cytokines (e.g., CCL2, IL-6, and IL-8) and pro-angiogenic factors (e.g., VEGF), and by promoting an invasive melanoma cell phenotype. All these features were found in the A375-RhS model, consistent with a more advanced stage of melanoma progression represented by this model.

## 4. Discussion

Progression and invasion of primary melanomas require the formation of a tumor-favorable TME to mediate the suppression of the anti-cancer responses of the immune system. This immune suppression is often accompanied by a CAF- and TGFβ-induced mesenchymal transition to a myofibroblast-like phenotype, which facilitates melanoma cell migration and invasion into the underlying dermis by breaking down the BM [19]. Angiogenesis becomes crucial in later stages for providing nutrients and oxygen for further melanoma progression and for directing melanoma cell intravasation into the blood or lymphatic circulation and subsequent metastasis. Indeed, vascular proliferation has been clinically associated with increased tumor aggressiveness and worse prognosis in melanoma patients [35]. In early trials, anti-angiogenic therapies showed poor improvement in the outcomes of traditional chemotherapy, but, with the advent of immune check-point inhibitors (ICIs), combination therapies with anti-angiogenic agents have gained interest and led to various clinical trials to pursue better outcomes in the treatment of advanced melanoma [35,41]. Little research has focused on developing 3D organotypic Mel-RhS to study human tumor-induced angiogenesis and only in recent years have ECs been integrated into a tissue-engineered 3D melanoma model [17].

Here, we describe the development of in vitro human melanoma models by integrating six melanoma cell lines (A375, COLO829, G361, MeWo, RPMI-7951, and SK-MEL-28) in a 3D skin environment. Melanoma cells mimicked early and later stages of local tumor progression and their accompanying tumor hallmarks in terms of cell growth and dermal expansion, immune modulation, and induction of angiogenesis and myofibroblast/CAF formation (see overview in Table 2). This dataset thus represents a significant advancement on our previous study focused on the immune suppressive properties of the SK-MEL-28 cell line in the skin model [15].

Whereas no nest formation could be observed when RPMI-7951 cells were integrated into the 3D model, COLO829-RhS and G361-RhS showed very limited melanoma nest formation after four weeks in air-exposed conditions. In line with this, no increased secretion of inflammatory cytokines or angiogenic factors was detected in their supernatants. Although some melanoma nest formation could be found in the case of models constructed with COLO829 and G361 cells, the lack of increased factors in the analyzed secretome (compared to the control 3D RhS model without melanoma) might suggest these cell lines are unable to induce the formation of a favorable TME and rather mimic very initial stages of melanomagenesis, which might also explain their very limited growth in the 3D model.

MeWo cell growth resembled the initial stages of melanoma progression without the higher secretion of immune modulatory cytokines (only a trend for M-CSF), but with higher secretion of a few pro-angiogenic factors (Flt-1 and VEGF). However, the released levels of these factors in the culture supernatants were not sufficient to induce EC sprouting in vitro. This might indicate that the formed tumor resembles more of an in situ/RGP-like phenotype rather than a transition towards VGP and eventually advanced melanoma. Although we do not yet clearly understand why this cell line seems to mimic very early stages of melanoma progression given its site of origin (i.e., lymph node), a possible explanation is that these melanoma cells underwent mesenchymal-to-epithelial transition (MET) to revert back from a mesenchymal to an epithelial state. This would have allowed them to proliferate at the metastatic site and develop into macro-metastases. This might also explain why this cell line is highly proliferative in the 3D model rather than invasive.

On the other hand, the previously described SK-MEL-28-RhS recapitulated the initial stages of melanoma invasion by showing a potential for inducing both an immune suppressive milieu via IL-10, M-CSF, and TGFβ release, as well as a pro-angiogenic environment, likely mediated by the release of Flt-1 and VEGF, the secretion of which was upregulated in the culture supernatants. While we previously demonstrated that the ability of SK-MEL-28-RhS-derived supernatants to skew monocyte differentiation away from DC differentiation and towards a suppressive M2-like macrophage phenotype was partly due to IL-10 [15], here we showed additional contributions by M-CSF and TGFβ. In vivo, IL-10, M-CSF, and TGFβ are all involved in the polarization and accumulation at the tumor site of tumor-associated macrophages (TAMs), which display features of alternatively activated M2 macrophages [42,43] and contribute to the suppression of anti-cancer T-cell-mediated immune responses [6,44,45,46]. Here, we found M-CSF and TGFβ secreted from SK-MEL-28-RhS to also play a role in the SK-MEL-28-RhS-mediated immune modulation, which might explain why IL-10 blockade alone was not sufficient to reverse the melanoma-induced conversion of monocyte into M2-like cells [15]. However, anti-M-CSF and anti-TGFβ were still insufficient (either alone or in combination with anti-IL-10) to fully counteract the skewing of monocytes towards M2-like CD14^+^CD163^+^CD16^+^ cells. This indicates the involvement of additional, as yet unidentified factors.

Melanoma-secreted TGFβ is also involved in the metabolic reprogramming of skin fibroblasts into CAFs, which can be identified by their α-SMA expression [47] and can aid melanoma progression by promoting immune escape and angiogenesis. This is consistent with previous literature reporting the role of TGFβ in influencing CAF differentiation by sustaining the increase in reactive oxygen species that modulate α-SMA expression [48]. Therefore, secreted TGFβ in this melanoma model may induce both immune suppression and CAF formation. The integration of SK-MEL-28 cells into the 3D RhS model indeed resulted in a clear population of α-SMA^+^ cells around the melanoma nests, suggesting the promotion of a melanoma-induced transition of fibroblasts into myofibroblast-like cells in their proximity. It has to be noted, however, that a small population of α-SMA^+^ fibroblasts was also present in the RhS controls, especially at the epidermal–dermal junction, which might have been due to the presence of fibrin in the hydrogel. Fibrin is naturally involved in wound healing and might have thus led to the creation of a basal inflamed-like state and fibroblast activation in the control RhS model.

Lastly, A375 displayed the most advanced stage of melanoma development by clear melanoma expansion into the reconstructed dermis and by the secretion of a plethora of (anti-)inflammatory cytokines (GM-CSF, IL-6, IL-8, IL-10, M-CSF, and TGFβ) involved in immune suppression and melanoma progression. Indeed, consistent with previous studies reporting elevated levels of the immune suppressive factors IL-10 [49,50,51] and M-CSF [52] in the blood of advanced melanoma patients, and similar to what we previously reported for SK-MEL-28-RhS [15], upregulated secretion of IL-10 and M-CSF was also found in A375-RhS-derived culture supernatants. These supernatants were indeed able to misdirect monocyte differentiation towards a M2-like subset (defined as CD14^+^CD163^+^CD16^+^), suggesting a potential for the formation of an immune suppressive skin microenvironment following A375 cell integration. However, not one of the analyzed cytokines (IL-6, IL-8, CXCL10, CCL2, CCL5, VEGF, IL-10, TGFβ, and M-CSF) correlated with the induction of this M2-like phenotype, suggesting the involvement of (a combination of) as yet unidentified factors. In this regard, it is important to note that, although removed prior to supernatant conditioning and collection, during RhS and Mel-RhS air-exposed culture, the medium contained hydrocortisone, an immune suppressive compound. This could have affected the overall inflammatory status of the RhS and Mel-RhS cultures.

Interestingly, TGFβ secretion by A375-RhS was highest among the Mel-RhS models. TGFβ levels were negatively correlated with Mel-RhS contraction, leading to a significant decrease in A375-RhS surface area, consistent with the presence, in the A375-RhS model, of α-SMA^+^ fibroblasts and their higher contractile abilities [53]. Besides TGFβ, also other factors, which were only upregulated in the A375-RhS model, such as IL-6 and IL-8 [37], might also have been involved in this A375-RhS-induced fibroblast activation. This would explain why SK-MEL-28-RhS, despite secreting considerable levels of TGFβ, did not induce contractility and is in line with the association of IL-6 and IL-8 with melanoma progression and metastasis [54,55]. IL-6 and IL-8 serum concentrations of melanoma patients have been found to be predictive of response to immunotherapies [56] and to correlate with poor prognosis [54]. IL-6 signaling has also been associated with resistance to BRAF inhibitors and immunotherapies [57,58,59], while IL-8 has been linked to the recruitment and activation to the TME of MDSCs, neutrophils, and other myeloid cell populations [60]. Elevated levels of these cytokines thus further support the hypothesis that the A375-RhS model represents a more advanced stage of melanoma development.

The integration of A375 cells in the in vitro RhS model also led to an increased secretion of the pro-angiogenic factors PlGF, Tie-2, Flt-1, VEGF, VEGF-C, and VEGF-D in culture supernatants, which were able to induce EC sprouting. Interestingly, higher secretion of VEGF-C or VEGF-D by A375-RhS suggests an ability to induce lymph angiogenesis, mediating lymphatic metastasis. This makes A375-RhS an attractive model to study lymph EC-tumor cell cross-talk and metastasis to the lymph nodes in a 3D in vitro human setting, e.g., with the aid of (multi-)organ-on-chip technology.

Melanoma is one of the most heterogeneous human cancers and this feature is indeed showcased by how highly variable the melanoma cell lines we used in this study were in terms of mutational status, invasive, immune suppressive, and angiogenic capabilities. With this heterogeneity in mind, we aimed to generate a 3D melanoma-in-skin model using different cell lines (i.e., different mutations and site of origin). While we could not find a clear correlation between melanoma cell growth and mutational status, as many different factors can come into play in determining cell survival, expansion, and ability to influence the surrounding environment, we have provided models that could mimic different stages of melanoma progression and could be thus further tailored for various purposes. For instance, immune cells could be integrated into the model to study their activation or suppression in situ, while the addition of blood or lymph ECs would aid to investigate melanoma cell migration to either the lymphatic or blood vessels. This would provide an attractive tool to further validate and expand the here reported data and investigate the different possible routes for melanoma immune editing and tumor cell dissemination in a fully human 3D setting.

While we chose to first characterize the in vitro model by incorporating commercially available cell lines, we developed a Mel-RhS which could be used to also test patient-derived melanoma cells in the future. Incorporating melanoma cells/cell lines with specific mutations/characteristics would give researchers the opportunity to test in a 3D setting the efficacy of drugs targeting a specific aspect of melanoma. The use of patient-derived specimen or low-passage bulk cultures would also make it possible to study these aspects in relation to intra-tumor heterogeneity, an important cause of therapy resistance. While of course our Mel-RhS still has limitations and is yet to fully recapitulate the complexity of the in vivo tissue microenvironment, it does demonstrate how such a model could be used to study and investigate melanoma features. In addition, the limited secretion of factors that we found from classical 2D melanoma cell monolayers, compared to that from the 3D models, highlights again the need for organotypic models to reflect the more complex structures of the human body.

## 5. Conclusions

Here, by investigating different melanoma cell lines in the reconstructed skin environment, we described six Mel-RhS models that can be used to study different stages of early and advanced in situ melanoma development in a 3D human in vitro setting. The presented Mel-RhS models serve as an attractive base for 3D cultures of human melanoma and provide ample opportunities to study growth, secretomics, CAF differentiation, and immune suppressive and angiogenic potential. Such models can be further optimized and tailored, for instance by incorporating additional cell types (e.g., immune cells or ECs), to obtain physiologically evermore relevant models to study specific mechanisms underlying melanoma development and progression. Moreover, while we chose to use well-characterized melanoma cell lines, the presented model can also be constructed by employing patient-derived melanoma cells, which would provide an extremely appealing platform in the context of personalized medicine.

## Figures and Tables

**Figure 1 cancers-15-02849-f001:**
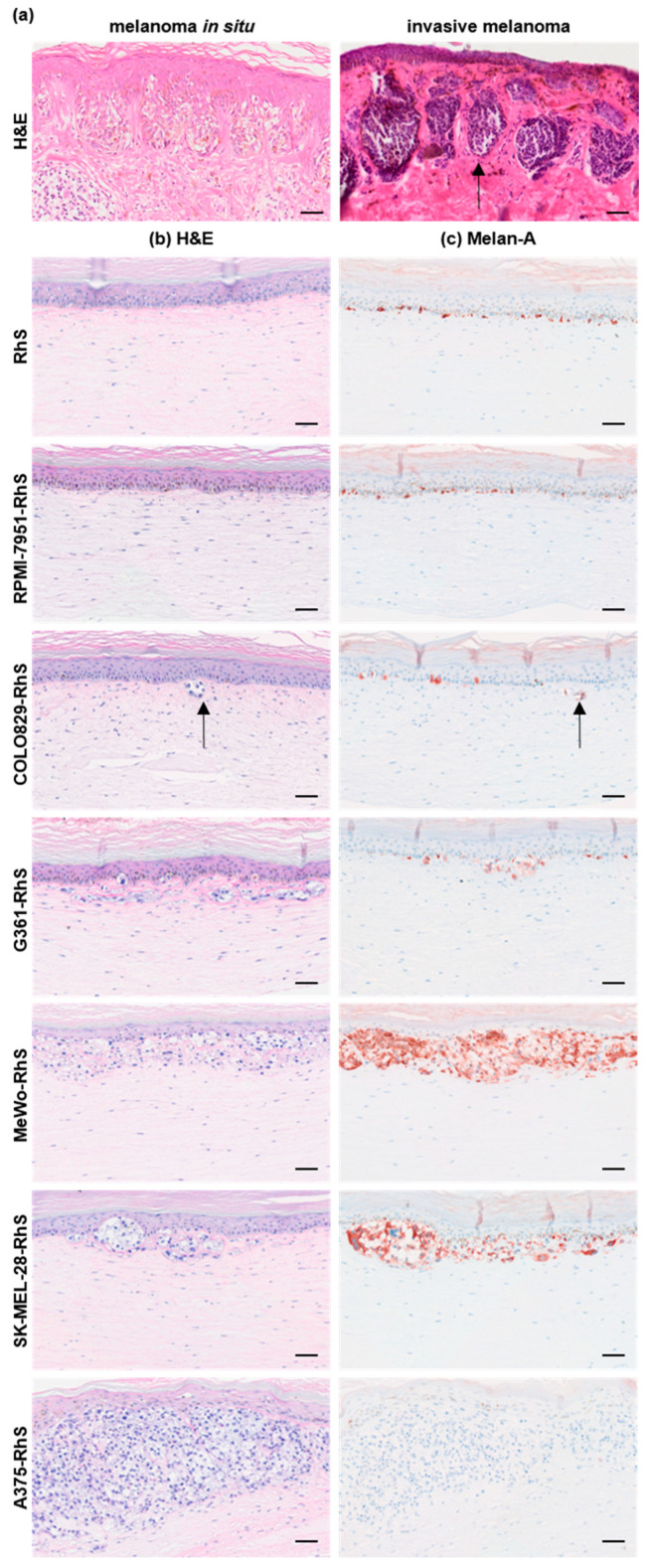
Growth and dermal spreading of melanoma cells in patients and in the in vitro models. (**a**) H&E shows melanoma growth and nest formation (see black arrow) in melanoma biopsies. RhS and Mel-RhS constructed with either A375, COLO829, G361, MeWo, RPMI-7951, or SK-MEL-28 melanoma cells were cultured for 4 weeks at the air–liquid interface and stained (**b**) by H&E and (**c**) for Melan-A. Melan-A^+^ healthy melanocytes were visible in the RhS, while Melan-A^+^ COLO829, G361, MeWo, and SK-MEL-28 cells could be observed at the epidermal–dermal junction of the melanoma model to different extents (indicated by black arrows for COLO829). A375 is an amelanotic cell line and therefore not positive for Melan-A. No visible RPMI-7951 nests could be detected. Representative pictures of H&E and Melan-A staining on FFPE 5 μm-thick tissue sections of at least four independent experiments, each with an intra-experiment replicate, are shown. Scale bar = 50 μm.

**Figure 2 cancers-15-02849-f002:**
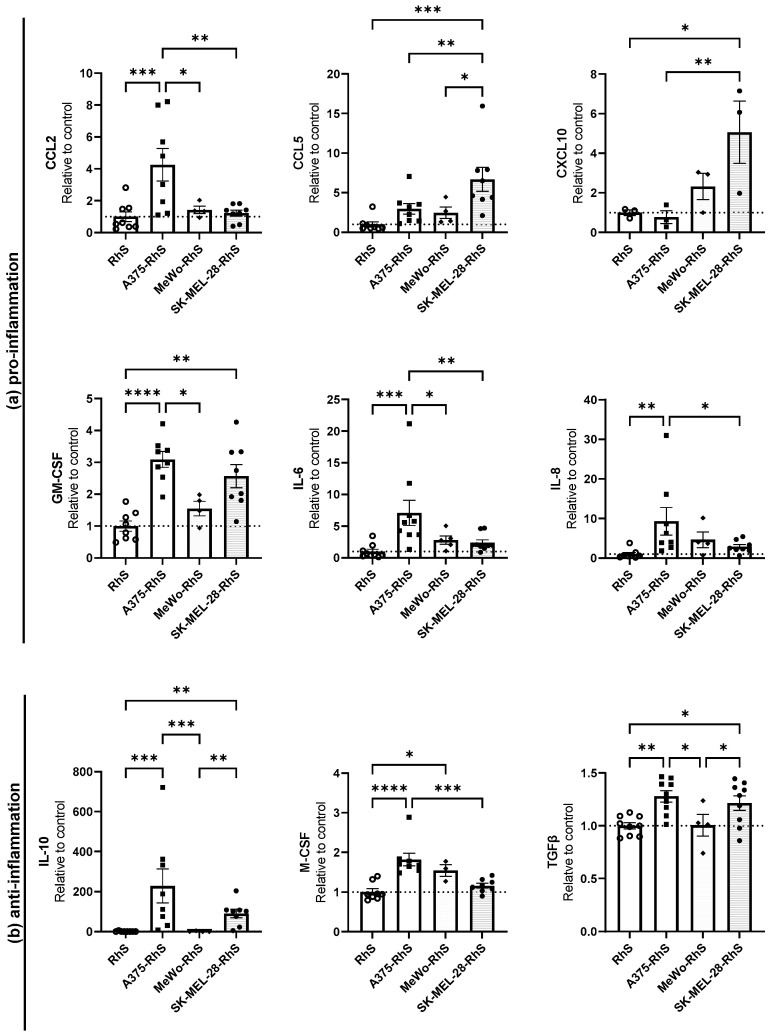
Cytokine secretion in the supernatants from either RhS (white circles) or Mel-RhS constructed with either A375 (black squares), MeWo (black diamonds), or SK-MEL-28 (black circles) cells. After 4 weeks at the air-liquid interface, medium was refreshed and culture supernatant was collected over a period of 24 h. Supernatants were analyzed for secretion of (**a**) pro-inflammatory and (**b**) anti-inflammatory factors by means of ELISA (GM-CSF, M-CSF, and TGFβ) or CBA (CCL2, CCL5, CXCL10, IL-6, IL-8, and IL-10). Results are shown as mean  ±  SEM (* *p*  <  0.05, ** *p* < 0.01, *** *p* < 0.001, and **** *p* < 0.0001; ordinary one-way ANOVA or Kruskal-Wallis test). Cytokine secretion in culture supernatants from at least three independent experiments, each performed with an intra-experiment replicate, is shown. Cytokine levels are normalized to the mean cytokine secretion of the control RhS, set as 1 (dotted line).

**Figure 3 cancers-15-02849-f003:**
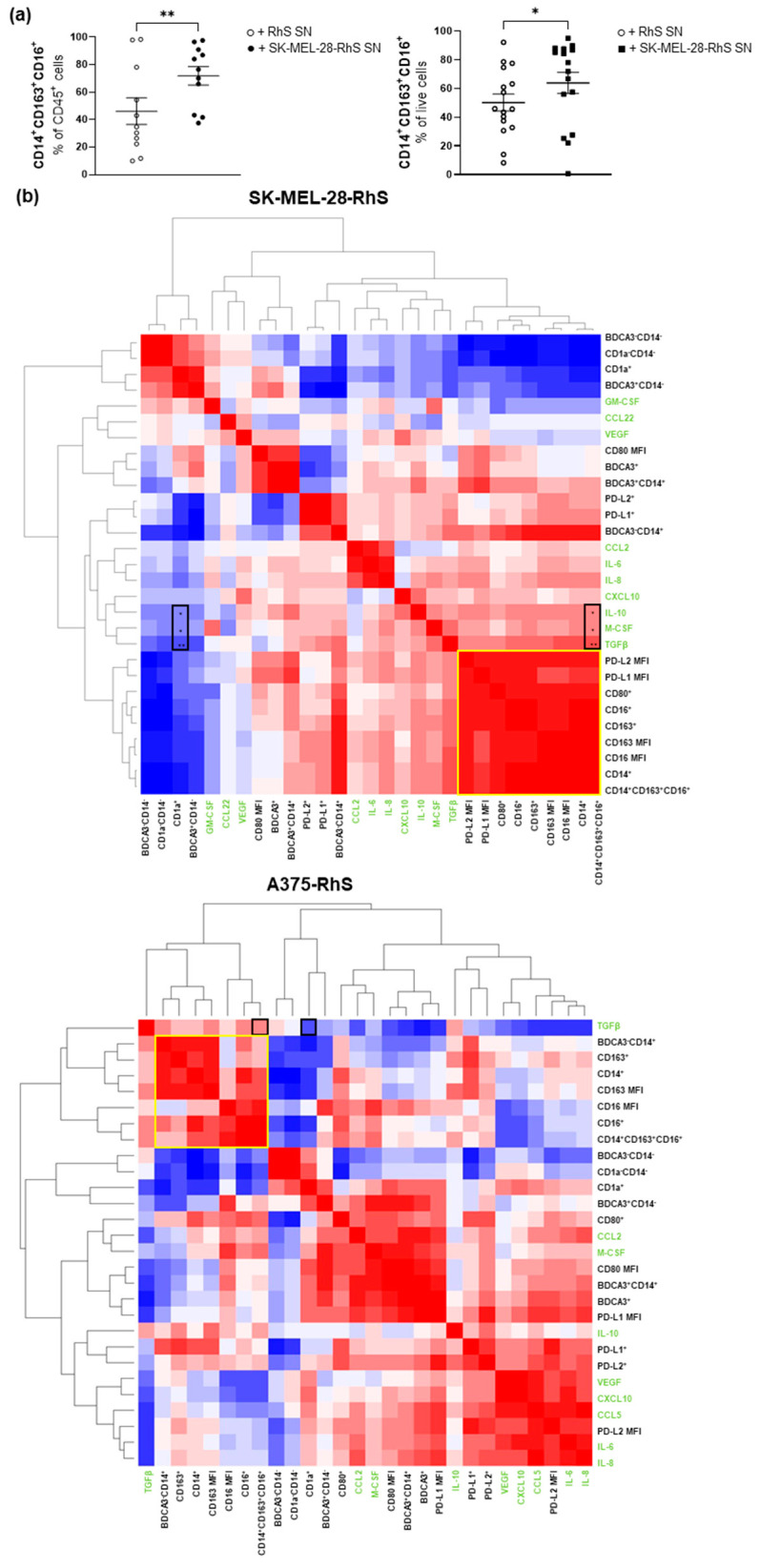
moDC phenotype after exposure to melanoma reconstructed human skin (Mel-RhS) culture supernatants for 6 days compared to its control (RhS) and its relation to cytokines levels secreted into culture supernatants. (**a**) Frequency in percentage of cells with an M2-like phenotype (defined as CD14^+^CD163^+^CD16^+^) exposed to culture supernatants from either RhS (+RhS SN; open circles), SK-MEL-28-RhS (+SK-MEL-28-RhS SN; closed circles, left graph), or A375-RhS (+ A375-RhS SN; closed rectangles, right graph). Results are shown as mean ± SEM (* *p*  <  0.05 and ** *p*  <  0.01; paired *t*-test). (**b**) Correlation matrix of cytokine levels (listed in green font) secreted from RhS and SK-MEL-28-RhS (top graph) or RhS and A375-RhS (bottom graph) and the expression of surface markers (in black font) on the monocytic population during monocyte-to-moDC differentiation upon exposure to the respective supernatants. Yellow boxes denote M2-like macrophage co-regulated clusters. Black boxes indicate correlation between the cytokines IL-10, M-CSF, and/or TGFβ and expression of CD1a and co-expression of CD14, CD163, and CD16. Significance levels of correlations: * *p* < 0.05 and ** *p* < 0.01. (**c**) Percentages of CD1a^+^_,_ CD14^+^, and M2-like cells (defined as CD14^+^CD163^+^CD16^+^) within the CD45^+^ monocytic cell population and their correlation with M-CSF and TGFβ levels secreted in the supernatants of RhS (open circles) or SK-MEL-28-RhS (closed circles). Results are shown with the 95% confidence bands of the best-fit line. Both *p*-value and Pearson *r* value are shown.

**Figure 4 cancers-15-02849-f004:**
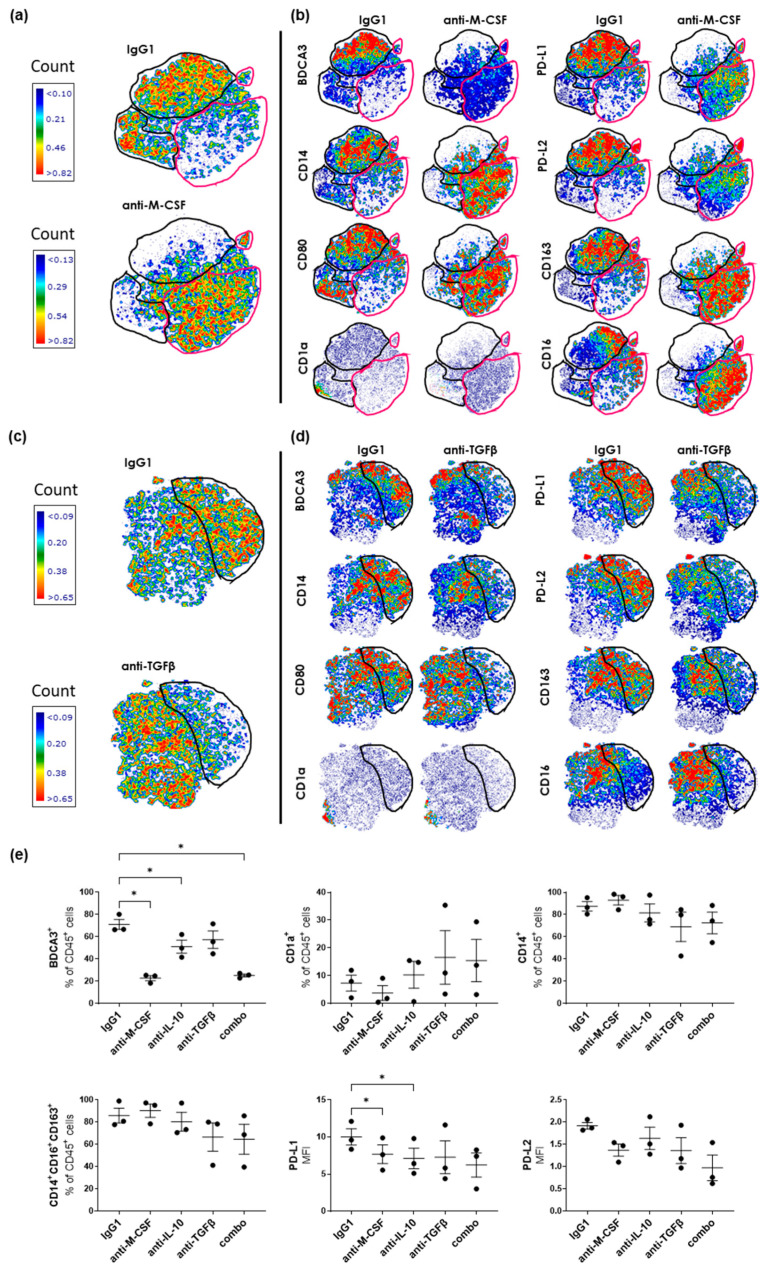
High-dimensional analysis of the phenotype of monocytes conditioned by supernatants derived from RhS or SK-MEL-28-RhS cultured in the presence or absence of M-CSF, IL-10, or TGFβ neutralizing antibodies. Differences in the t-SNE analyses between IgG1 and (**a**) anti-M-CSF or (**c**) anti-TGFβ conditions. (**b**,**d**) Differences between IgG1 and the respective neutralizing antibody in the intensity and the distribution of expression of BDCA3, CD14, CD80, CD1a, PD-L1, PD-L2, CD163, and CD16 in the t-SNE analysis. The same gates as a and c are depicted between the IgG1 and the respective neutralizing antibody conditions in b and d. Data derived from three sets of supernatants and one monocyte donor. (**e**) Frequency of BDCA3^+^, CD1a^+^, CD14^+^, and CD14^+^CD163^+^CD16^+^ cells and geometric mean intensity (MFI) of PD-L1 and PD-L2 in the CD45^+^ cells (i.e., monocytes cultured in the presence of the DC differentiation-inducing cytokines GM-CSF and IL-4) after incubation with Mel-RhS-derived supernatant pre-treated with either IgG1, anti-M-CSF, anti-IL-10, anti-TGFβ, or the combination of all three neutralizing antibodies (combo) (N  =  3; mean  ±  SEM is shown; * *p*  <  0.05; ordinary one-way ANOVA). Black gates denote highest density population in IgG1, magenta gates in M-CSF conditions.

**Figure 5 cancers-15-02849-f005:**
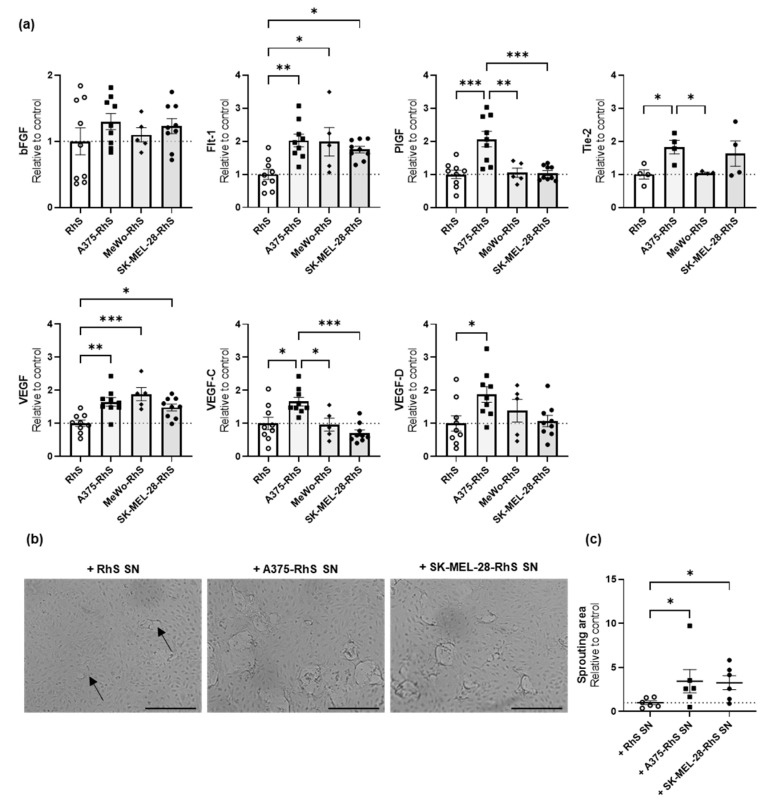
Angiogenic potential of RhS, A375-RhS, MeWo-RhS, and SK-MEL-28-RhS models. (**a**) Levels of pro-angiogenic factors in RhS- and Mel-RhS-derived culture supernatants constructed with either A375, MeWo, or SK-MEL-28 cells. After 4 weeks at the air–liquid interface, medium was refreshed, and culture supernatant was collected over a period of 24 h and analyzed for secretion of bFGF, Flt-1, PlGF, Tie-2, VEGF, VEGF-C, and VEGF-D. Results are shown as mean ± SEM (* *p*  <  0.05, ** *p* < 0.01, and *** *p* < 0.001; ordinary one-way ANOVA or Kruskal-Wallis test). Data from at least four independent experiments performed in duplicate are shown. Secreted cytokine levels were normalized to the mean cytokine secretion of the control RhS, set as 1 (dotted line). (**b**) Surface view of EC sprouting in a fibrin matrix after stimulation for 48 h with 30% RhS-, A375-RhS-, or SK-MEL-28-RhS-derived culture supernatants. Black arrows indicate examples of EC sprouts. Scale bar = 250 μm. SN: supernatant. (**c**) Quantification of EC sprouting in response to exposure to RhS-, A375-RhS-, or SK-MEL-28-RhS-derived culture supernatants. The amount of sprouting was calculated as the surface area of the sprouts as a percentage of the total surface. The sprouting area was normalized to the control, i.e., ECs stimulated with RhS-derived culture supernatants, set as 1 (dotted line). Supernatants from six independent experiments performed in duplicate were used with ECs from two different donors. Results are shown as mean  ±  SEM (* *p*  <  0.05; Kruskal-Wallis test). SN: supernatant.

**Figure 6 cancers-15-02849-f006:**
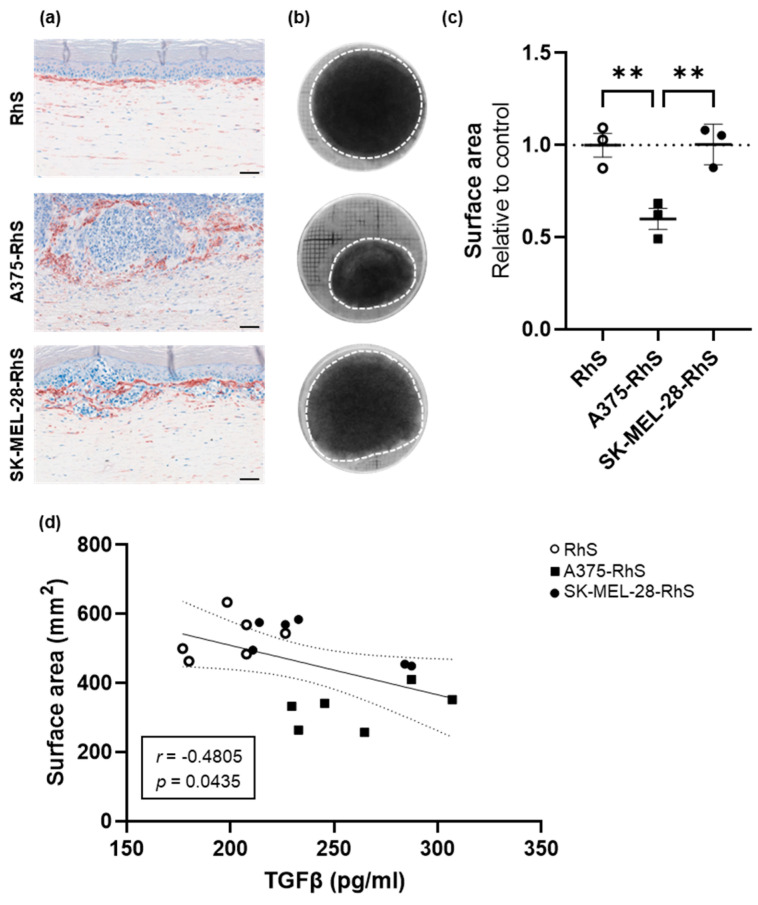
The presence of A375 melanoma cells in the RhS model leads to contraction: evidence of myofibroblast formation. (**a**) Representative pictures of α-SMA staining on FFPE sections of RhS, A375-RhS, and SK-MEL-28-RhS. Clear α-SMA expression can be observed around the melanoma nests formed by A375 and SK-MEL-28 cells. Scale bar = 50 μm. (**b**) Representative photos of RhS, A375-RhS, and SK-MEL-28-RhS models after 4 weeks at the air–liquid interface. White dotted line represents the edges of each RhS culture. (**c**) After 4 weeks in air-exposed conditions, the surface area of RhS, A375-RhS, and SK-MEL-28-RhS was measured. The incorporation of the A375 cell line into the reconstructed skin model led to a significant decrease in the surface area compared to both SK-MEL-28-RhS and RhS. No difference between RhS and SK-MEL-28-RhS could be observed. The surface area was normalized to the mean area of the RhS controls (dotted line). Results from three independent experiments, each performed with an intra-experiment replicate, are shown as mean  ±  SEM (** *p* < 0.01; ordinary one-way ANOVA). (**d**) The surface area after 4 weeks of air-exposed culture and correlation with the TGFβ levels secreted in the supernatants derived from RhS (white circles), A375-RhS (black squares), or SK-MEL-28-RhS (black circles). Results are displayed with the 95% confidence bands of the best-fit line. Both *p*-value and Pearson *r* value are shown.

**Table 1 cancers-15-02849-t001:** Genotypic features and origin of the melanoma cell lines used in this study. Where no reference is reported, information was acquired directly from the respective suppliers’ website.

Cell Line	BRAF Status	PTEN Status	NRAS Status	Origin	Supplier
A375	c.1799T>A	WT [27]	WT [27]	skin	ATCC
COLO829	c.1799T>A	c.493_634del142	WT	skin	ATCC
G361	c.1799T>A			skin	ATCC
MeWo	WT [28]	WT [29]	WT [30]	lymph node	ATCC
RPMI-7951	c.1799T>A	c.1_79del79	WT [30]	lymph node	ATCC
SK-MEL-28	c.1799T>A [28,31]	A499G [27]	WT [27,30]	lymph node	CLS Cell Lines Service GmbH

**Table 2 cancers-15-02849-t002:** Overview of the developed melanoma models and their respective features in terms of immune modulatory and angiogenic potential. Models are displayed in order of increased invasive potential.

Mel-RhS	Melanoma Stage	Cytokines	Immune Modulation	Angiogenic Factors	Sprouting
RPMI-7951-RhS	None	None	N.D.	None	N.D.
COLO829-RhS	Very early stage	None	N.D.	None	N.D.
G361-RhS	Very early stage	None	N.D.	None	N.D.
MeWo-RhS	RGP	None	N.D.	↑ Flt-1, VEGF	No
SK-MEL-28-RhS	Early invasive stage	↑ CCL5, CXCL10, GM-CSF, IL-10, TGFβ	Via IL-10, M-CSF, TGFβ	↑ Flt-1, VEGF	Yes
A375-RhS	Late invasive stage	↑ CCL2, GM-CSF, IL-6, IL-8, IL-10, M-CSF, TGFβ	Needs to be further investigated	↑ Flt-1, PlGF, Tie-2, VEGF, VEGF-C, VEGF-D	Yes

RGP: radial growth phase; ↑ = upregulated; N.D. = not determined.

## Data Availability

The data are not publicly available due to privacy.

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
