# Peer review of "A Reconstructed Human Melanoma-in-Skin Model to Study Immune Modulatory and Angiogenic Mechanisms Facilitating Initial Melanoma Growth and Invasion"

_cancers, 2023, doi:10.3390/cancers15102849_

Round 1
Reviewer 1 Report
This is a well-designed and well written manuscript.
The only problem I have is the design of Figure 2. The symbols are to big, I recommend to redesign this figure.
Author Response
Please see the attachment. Line numbers refer to viewing the document under the All Markup option for track changes.

Reviewer 2 Report
In this work authors described the set up of in vitro 3D model of melanoma skin already characterized for SK-MEL-28. The authors used different cell lines to establish the skin model and they characterized only few type analyzing different features like inflammatory factors, angiogenesis and immunosuppressor abilities. the work is not new as it has already been applied previously by the same authors. However they demonstrated the applications of the model to different oncological fields. I suggest the publication of the work after few minor corrections:
-Table S1. the table is difficult to understand due to the large amount of data, maybe is simpler to add fold-change instograms as presented in Figure 2. Moreover the authors could better underline or explain a possible correlation between the lack of nest formation in COLO829-RhS, G361-RhS, and RPMI-7951-RhS models and the analysis of inflammatory factors.
-Figure 3 is really crowded. Labels in the graphs are hardly readable, especially figure 3a.
-There is a mismatch in file names of supplementary figures, please check figures in the appendix and figures in the separate supplementary file.
-In figure 5, How did the authors choose the type of melanoma model to use? because they have also added the MeWo-RhS model for the vascularization? and why MeWo model is not analyzed in 5b and c.
-"Levels of α-SMA expression were clearly higher than in that of RhS (Figure 6a), COLO829-RhS, G361-RhS, MeWo-RhS, or RPMI-7951-RhS (Figure S2), which fits with the more advanced invasive states within A375-RhS and SK-MEL-28-RhS compared to the other cell line models." This phrase is not clear. Please describe better the aim and the results that you want to show in this part.
Author Response

(The authors gave the same response as above.)

Reviewer 3 Report
Dear Authors,
Thank you for preparing this interesting manuscript on the reconstructed human melanoma-in-skin model. As each year melanoma incidence and mortality raise, and melanoma is one of those cancers which quickly gains resistance to treatment, investigations seeking to understand how this cancer works are very important and necessary. Especially important is carrying out basic research, like this one where the Authors have recreated 3D human skin with introduced melanoma cells for various applications in research. The article was well-planned and quite well-written. Although with most results and conclusions I agree, some points require clarifications and deeper discussion.
Major concerns:
1. In this study 6 commercial cell lines were used. These cell lines are from different origins, for example, MeWo cell line was derived from the metastasis to the lymph node, whereas G361 cell line is a skin-derived metastatic melanoma. Taking this into account and based on the received results within each Mel-RhS model please comment on these relations in the discussion part.
2. In terms of the pro- and anti-angiogenic factors secretion, it would be good to check and compare with each of the created models, if the used cell lines are able to produce these factors when cultured alone and on what level. If they can produce them in a monoculture and they stopped producing them in these models then this is a discovery. However, these cell lines may have generally limited production of various factors thus in these models they are also not able to produce it. This seems extremely important (in my opinion) in the main discovery of the Authors in the content of MeWo cell line, which is derived from lymph metastasis, but within these models, it resembles more of an in situ/RGP-like phenotype.
3. Melanoma is one of the most heterogeneous human cancers, which may be the main cause of drug resistance development. To what extent do the created models reflect this research problem? Please comment on this topic as well.
4. This next comment relates to subsections 2.2.1 and 2.2.3. Although it is normal that the Authors decide not to include full procedures if they have published them earlier and reference them with the note "as previously described", however, it would be good to reference the article with the full description allowing the recreation of a procedure by a reader. With reference 26, it is possible to recreate the keratinocyte isolation procedure, but reference 25 is insufficient for the fibroblast isolation. It also has a reference to a previous work "as described previously [34]", thus it seems that here the Authors did not cite the correct source. With reference 27, it is not possible to recreate the ECs isolation procedure either, again the Authors did not cite the correct source (in reference 27 they claim "as described previously [29]"). Please correct this.
5. A similar situation was found in subsection 2.3 (Mel-RhS was constructed as previously described [15]), the Authors did not cite the correct source as reference 15 claims "RhS was constructed essentially as previously described [17]". Please correct this.
6. The reference style was selected in such a way that it is hard to estimate the level of excessive self-citation by the Authors.
Minor suggestions:
· - please correct grammar in lines 33-36, 261-262, 466-467, 578-583, 615-616
· - figure 2a is too small (low readability)
· - graphs in figure 5a are too small
· - please unify the abbreviation (Mel-RhS or (Mel-)RhS); (α-SMA or αSMA); (TGFβ or TGF-β in the whole manuscript
· - line 486, please correct "Results from three independent experiments"
- line 565, please correct (anti-MCF) -> anti-M-CSF
With kind regards
Author Response

(The authors gave the same response as above.)

Round 2
Reviewer 3 Report
Dear Authors,
Thank you for revising your manuscript. I am quite satisfied with most of your corrections and provided answers.
However, considering my question no. 2 and the data that you have presented (release of angiogenic factors in the cell supernatant) it is clear that the situation is exactly what I have described (cell lines have limits in the production of various factors in the laboratory conditions). I strongly believe that this should be pointed out in your manuscript. Please prepare a figure or table where the reader could clearly compare these data from cell culture (A375 and SK-MEL-28 cells) and RhS models (A375-RhS and SK-MEL-28-RhS), and comment on the problem in the manuscript. These data raise a serious problem that is observed with various models attempting to reflect the more complex structures of the human body.
With kind regards
